# The Role of Brain-Derived Neurotrophic Factor (BDNF) in Diagnosis and Treatment of Epilepsy, Depression, Schizophrenia, Anorexia Nervosa and Alzheimer’s Disease as Highly Drug-Resistant Diseases: A Narrative Review

**DOI:** 10.3390/brainsci13020163

**Published:** 2023-01-18

**Authors:** Aleksandra Gliwińska, Justyna Czubilińska-Łada, Gniewko Więckiewicz, Elżbieta Świętochowska, Andrzej Badeński, Marta Dworak, Maria Szczepańska

**Affiliations:** 1Department of Pediatrics, Faculty of Medical Sciences in Zabrze, Medical University of Silesia, 40-055 Katowice, Poland; 2Department of Neonatal Intensive Care, Faculty of Medical Sciences in Zabrze, Medical University of Silesia, 40-055 Katowice, Poland; 3Department of Psychiatry, Faculty of Medical Sciences in Zabrze, Medical University of Silesia, 40-055 Katowice, Poland; 4Department of Medical and Molecular Biology, Faculty of Medical Sciences in Zabrze, Medical University of Silesia, 40-055 Katowice, Poland; 5Department of Pediatric Nephrology with Dialysis Division for Children, Independent Public Clinical Hospital No. 1, 41-800 Zabrze, Poland

**Keywords:** brain-derived neurotrophic factor, epilepsy, depression, schizophrenia, anorexia nervosa

## Abstract

Brain-derived neurotrophic factor (BDNF) belongs to the family of neurotrophins, which are growth factors with trophic effects on neurons. BDNF is the most widely distributed neurotrophin in the central nervous system (CNS) and is highly expressed in the prefrontal cortex (PFC) and hippocampus. Its distribution outside the CNS has also been demonstrated, but most studies have focused on its effects in neuropsychiatric disorders. Despite the advances in medicine in recent decades, neurological and psychiatric diseases are still characterized by high drug resistance. This review focuses on the use of BDNF in the developmental assessment, treatment monitoring, and pharmacotherapy of selected diseases, with a particular emphasis on epilepsy, depression, anorexia, obesity, schizophrenia, and Alzheimer’s disease. The limitations of using a molecule with such a wide distribution range and inconsistent method of determination are also highlighted.

## 1. Introduction

In the 1950s, a new era in the development of neurology and psychiatry began when Levi-Montalcini and Hamburger discovered the first neurotrophin, nerve growth factor (NGF) [1]. However, it was not until three decades later, in 1982, that Barde et al. isolated brain-derived neurotrophic factor (BDNF) from the pig brain, and their discovery shed new light on the understanding of brain processes [1,2]. The neurotrophin family also includes neurotrophin 3 (NT3) and neurotrophin 4/5 (NT4/5), also known as neurotrophin 4 or 5 [3,4,5]. Neurotrophins are synthesized as immature forms that have a high affinity for the p75 neurotrophin receptor (p75NTR) and low affinity for tropomyosin-related kinase (Trk) receptors. They are then converted by proteolysis into mature forms of neurotrophins that act through Trk receptors, although they also have a low affinity for p75NTR [5,6]. Whereas NGF binds with high affinity to TrkA receptors, BDNF and NT4/5 bind to TrkB receptors and NT3 binds to TrkC receptors, also TrkA and TrkB receptors, which are less potent [5,6,7] (Figure 1).

Neurotrophins regulate neuron survival, differentiation, and specification during embryogenesis and after birth [8]. However, in the mature nervous system, they modulate synaptic transmission and axonal morphology and ultimately affect behavior, learning, memory, cognition, and many other processes, some of which remain unexplored [4,6,8]. BDNF exerts neuroprotective and neuroactivating properties on excitatory and inhibitory neurons, with it being produced exclusively in excitatory neurons [9]. The initial focus was on the effects of BDNF on the central nervous system (CNS), and its greatest abundance was found in the hippocampus and frontal cortex [10]. Further studies have shown that BDNF is also present in peripheral organs such as the heart, intestine, thymus, spleen, lung, and kidneys [3,11,12].

The BDNF gene In humans has been mapped to chromosome 11p14.1. It consists of 11 exons at the 5′ end and is regulated by nine functional promoters, although the number of exons is 12 according to the National Library of Medicine (Gen ID) [5,13].

BDNF gene expression is regulated by endogenous and exogenous stimuli, e.g., stress, brain injury, physical activity, diet, and drugs, especially antidepressants [3,14]. In turn, in neonates, a relationship between BDNF levels and cigarette and alcohol consumption during pregnancy, as well as obesity in pregnant women, has been demonstrated [8].

BDNF is synthesized as proneurotrophin (pro-BDNF), with a molecular weight of 32–35 kDa, which can be cleaved by endoproteases (intracellular) or metalloproteinases (extracellular) into the mature form (mBDNF), with a molecular weight of 13–14 kDa, and is secreted as a mixture of pro-BDNF and mature BDNF, which differ in their affinity for receptors [6]. TrkB has an extracellular domain with glycosylation sites, a transmembrane segment, and an intracellular domain characterized by Trk activity [6]. TrkB activation triggers a signal, including phospholipase C gamma (PLC-γ), which activates protein kinase C (PKC), the mitogen-activated protein kinase (MAPK) pathway, which leads to the activation of various downstream effectors, and phosphatidylinositol 3-kinase (PI3-K), which activates the serine/threonine kinase AKT, also known as protein kinase B [15,16] (Figure 2). Mature BDNF is generally believed to be involved in protein synthesis, axonal growth, dendritic cell maturation, and synaptic plasticity, which can be defined as activity-dependent changes in the strength or efficacy of synaptic transmission at pre-existing synapses and are critical for learning and memory [16,17,18]. Pro-BDNF, meanwhile, has apoptotic effects via p75NTR [16,17]. In addition, there are differences in the way these receptors bind depending on the type of neurotrophin and the site at which it acts. For example, cholinergic neurons in the forebrain respond strongly to NGF and weakly to BDNF, whereas motor neurons in the spinal cord respond strongly to BDNF and weakly to NGF [4].

Of the neurotrophins mentioned above, BDNF has received the most attention because of its possible role in the development of numerous neurological and psychiatric disorders and because of its potential therapeutic value. However, because of its lack of specificity, its utility may be limited to monitoring responses to treatment without the possibility of using it as a predictive marker of disease onset.

Particularly problematic factors in describing BDNF-dependent processes are the method of its determination and the attempt to correlate central and peripheral BDNF concentrations, as studies provide different, sometimes contradictory, conclusions. Research has proven that the BDNF concentration is high in human serum and much lower in plasma [19]. More than 90% of BDNF found in blood is contained in platelets, which contrary to earlier assumptions that BDNF circulating in blood is mainly shipped with the CNS thanks to its ability to cross blood barriers, with secondary BDNF now thought to originate from platelets that release it upon platelet activation [19,20]. The peripheral BDNF concentration is usually measured in blood samples using enzyme-linked immunosorbent assays (ELISA) [21]. Despite confounding factors, serum BDNF levels are more stable than plasma concentrations and are therefore more frequently recommended [19]. Other authors point out that serum processing may result in the release of BDNF contained in platelets, which in turn confounds its correlation with BDNF contained in the CNS [21]. Nevertheless, many studies suggest that BDNF concentration in serum is strongly correlated with CNS concentration. For this reason, the measurement of peripheral BDNF levels is a suitable indicator of central BDNF expression [9,22]. Moreover, there is a negative correlation between serum BDNF stored in platelets and depression in humans [23]. The release of BDNF from platelets may be impaired in depressed patients, whereas antidepressants increase the release of BDNF from platelets, suggesting that BDNF from platelets is a factor contributing to the interaction between peripheral BDNF levels and depression [24,25].

Consistent with other studies, Glud et al. demonstrated a dependence between peripheral BDNF levels and gender, with serum BDNF levels being 25% higher in women compared with men, confirming that circulating BDNF is gender dependent. It has been previously shown that women have higher levels of BDNF expression in different brain regions [26].

Since the discovery of neurotrophins, especially their anti-apoptotic properties, the possibility of their therapeutic use has been sought, both in neurodegenerative and other neurological diseases and in psychiatric disorders. The first clinical studies in cell cultures and animal models reached different conclusions, pointing to the risk of side effects and the need for a more detailed understanding of their pharmacokinetic properties. A major challenge in the therapeutic use of neurotrophins remains the determination of the effective dose and the exact injection site within a well-defined time period [4].

This review focuses on the role of BDNF in highly drug-resistant diseases and highlights the hopes associated with its use in the diagnosis, therapy monitoring, and treatment of neuropsychiatric diseases.

## 2. Function of BDNF in the Nervous System

Because BDNF and TrkB are found primarily in the CNS, most diseases associated with the impaired expression of BDNF involve the field of neurology and psychiatry. In both rats and humans, high levels of BDNF have been detected in the hippocampus, amygdala, cerebellum, and cerebral cortex, with the highest BDNF and TrkB protein immunoreactivity being detected in the hippocampus [1,14].

For years, until the early 20th century, the process of neurogenesis was thought to be exclusive to the prenatal period. A breakthrough in neurobiology was the discovery of the proliferative potential of the mature mammalian brain, demonstrated during the research of Joseph Altman et al. in 1965 [27]. It is now recognized that new neurons are generated in certain brain structures throughout life, with only the rate of their proliferation and survival decreasing with age [28]. There are two areas where the continuous process of neurogenesis takes place. The area with the highest number of active cell divisions is the subventricular zone (SVZ), located near the lateral ventricles of the brain, and the second area is the subgranular zone (SGZ) in the dentate gyrus of the hippocampus. The control of multiplying of proliferatively active neural stem cells (NSCs) depends, among other things, on BDNF-dependent regulation [28,29]. In a study conducted by Pencea et al., BDNF was shown to promote the survival and differentiation of postnatal SVZ progenitor cells in vitro and to increase the number of neurons formed in the adult rostral migratory stream and olfactory bulb in vivo [30].

It is generally accepted that BDNF is involved in synaptic plasticity by regulating presynaptic and postsynaptic transmission. BDNF is known to be required for the proper development and maintenance of dopaminergic, GABAergic, cholinergic, and serotonergic neurons [15]. The first studies confirming the effects of BDNF on synaptic transmission were performed on Xenopus cultures [1]. Presynaptic BDNF signaling affects neurotransmitter release, whereas postsynaptic BDNF signaling is involved in the activity of ionotropic channels, e.g., α-amino-3-hydroxy-5-methyl-4-isoxazolepropionic acid receptor (AMPA-R), potential cation channels, and sodium and potassium channels [15].

Regarding the expression of TrkB in both principal neurons and interneurons, BDNF mRNA is detectable in principal cells but only at low or undetectable levels in interneurons. Interneurons mostly use gamma-aminobutyric acid (GABA), which is an inhibitory neurotransmitter, as their main, classical transmitter [31]. BDNF has a positive effect on the development of GABAergic synapses, but its effect in the mature brain on GABA-dependent transmission may be multidirectional. These relationships were described in the work of Porcher et al., which showed that BDNF may have a differential effect on GABAergic synaptic inhibition depending on factors such as: the stage of neuronal development, the function of a particular area of the brain, or the rate of the delivery of exogenous BDNF. The authors of the study believe that the seemingly opposite effect of BDNF in this regard may contribute to the maintenance of the homeostasis of GABAergic transmission [32].

## 3. Polymorphisms of BDNF Gene and Receptors

A dependence on polymorphisms in both the BDNF gene and related receptors, which may increase the risk of developing neuropsychiatric disorders, has been described in numerous studies [33,34,35,36,37]. A particular role is played by the functional single nucleotide polymorphism (SNP) in the BDNF gene, which consists of the substitution of adenine by guanine (A/G) at position 196 in exon 5 of the BDNF gene, resulting in the replacement of the amino acid valine (Val) by methionine (Met) in codon 66 (Val66Met, also known as Rs6265) [5,38].

It is estimated 25% to 50% of individuals in different populations have methionine at position 66 of the BDNF protein [20]. Changes in the structure modify the intracellular transport of pro-BDNF and BDNF secretion, leading to hippocampal dysfunction [36]. The Val66Met polymorphism has been shown to reduce synaptic plasticity and correlate with a higher risk of cognitive impairment in Parkinson’s disease [39]. On the basis of meta-analyses, a large group of Val66Met polymorphisms were found to have no effect on the risk of major depressive disorder (MDD), whereas a smaller number of patients (n = 1.002) showed an association between the polymorphism and the development of depression in later life [38,40]. Regarding Alzheimer’s disease, there are conflicting reports, but one of the recent reports in JAMA Neurology indicates that in dominantly inherited Alzheimer’s disease (DIAD), the clinical stage of disease and BDNF Met66 are associated with cognitive impairment and the level of site-specific tau phosphorylation, which plays a key role in the pathophysiology of Alzheimer’s disease [37]. Studies on temporal epilepsy demonstrate the protective role of the Val66Met polymorphism in the development of this disease [2,41].

Terracciano et al., in a meta-analysis of Val66Met and genome-wide association study (GWAS), found that the serum BDNF level was not dependent on Val66Met. They also pointed out that their study and other large research studies had found no association between Val66Met and the personality trait of neuroticism, mood disorders, ADHD, schizophrenia, and Alzheimer’s disease. In the same study, they draw attention to the SNP within and near the neurotrophic receptor tyrosine kinase 3 (NTRK3) gene, which encodes the TrkC receptor that binds neurotrophin 3 but not BDNF, but its polymorphism correlates with serum BDNF levels, which is likely the result of feedback on BDNF expression and storage [33].

Other studies in people with schizophrenia found that carriers of the Met allele had lower peripheral blood BDNF concentrations, which contradicts other studies that found no differences between Met carriers and Val/Val homozygotes in the measurement of peripheral BDNF in plasma and serum [33,42].

In addition, the association between late-onset Alzheimer’s disease and the C270T polymorphism in the BDNF gene and the association between the increased risk of suicide attempts in depressed patients and polymorphism of TrkB and p75NTR receptors have been described [5].

## 4. Clinical Implications

### 4.1. Epilepsy

Epilepsy is a chronic disorder of the central nervous system that affects an estimated 0.76% of the world’s population [43]. Except for a few established causes, the origin of epilepsy in the majority of patients cannot be determined by the currently available diagnostic methods.

The most common clinical form of epilepsy is temporal lobe epilepsy (TLE), which is characterized by a relatively high resistance to pharmacotherapy, often requiring surgical treatment to remove the epilepsy focus [29].

Patients with epilepsy suffer from mood and cognitive changes in addition to seizures or focal symptoms [2]. Despite advances in pharmacotherapy to reduce seizure frequency, there is still a group of roughly 15–30% of patients who have little or no response to conventional antiepileptic therapy, and it is in this group that hopes are pinned on the use of BDNF-dependent mechanisms [2,3].

Seizures are associated with the increased expression of transcription factors, neuropeptides, and growth factors [44,45,46]. An initial study by Gall and Isackon in 1989, showing that limbic seizures increase NGF mRNA levels, motivated further research to clarify the relationship between neurotrophin prevalence and epileptogenesis [2,47]. Later observations showed that BDNF, but not NGF, increased the frequency of excitatory postsynaptic miniature currents in Xenopus cultures [2]. There are studies showing an increase in BDNF gene expression in the hippocampus and temporal cortex in patients with temporal lobe epilepsy [29]. It remains unclear whether the excitatory action of BDNF is mainly presynaptic (e.g., by increasing glutamate release) or postsynaptic (e.g., by phosphorylating neurotransmitter receptors); in either case, its action is to reduce inhibitory transmission by gamma-aminobutyric acid (GABA) and to enhance excitatory glutamate-dependent synapses [2].

There are conflicting reports of changes in BDNF levels in both CNS and serum in patients with certain forms of epilepsy. In three different studies cited by Iughetti et al., higher serum BDNF levels were shown to be associated with a more severe course of TLE and other forms of primary epilepsy. At the same time, Iughetti et al. noted that the acute injection of BDNF into the brains of mice induces seizures that can be eliminated by BDNF blockade, while the chronic infusion of BDNF is inversely associated with decreased neuronal excitability, likely via multiple mechanisms, including an increase in central levels of neuropeptide Y (NPY) [2]. Despite numerous studies suggesting a proepileptogenic effect of BDNF, the meta-analysis by Nowroozi et al. showed no statistically significant differences in BDNF levels between patients with epilepsy and control subjects, although patients with partial epilepsy, in contrast to patients with primary generalized epilepsy, had lower BDNF levels than control subjects [41]. In addition, patients with temporal lobe epilepsy were found to have higher BDNF levels. However, the authors pointed out that there are several factors that may lead to erroneous conclusions from the analyzed studies [41]. On the one hand, the weakening of neuroprotection by limiting the effect of BDNF in the CNS may promote epileptic seizures. However, we may observe a secondary effect of seizure-related stress that may lead to a decrease in the BDNF concentration. This is suggested by the study by Lafrance Jr. et al., which showed a decrease in BDNF levels in patients who experienced a seizure, regardless of the underlying disease [48]. In addition, there are factors that influence the BDNF concentration, such as gender or age, which requires further research, but these have not always been taken into account [41]. Differences in in terms of BDNF concentration depend on the time of sampling, with the level being higher in the morning and remaining at a constant low level through the rest of the day in patients treated with antiepileptic drugs [49]. The authors of the meta-analysis point out that the use of antiepileptic drugs may be a major obstacle to correct conclusions in the vast majority of patients studied. Antiepileptic drugs downregulate BDNF production in both animals and humans. In addition, the difference in treatment regimens (generalized epilepsy vs. partial epilepsy) may have confounded the conclusion that BDNF levels were lower in partial epilepsy than in the healthy control group, which was not observed in generalized epilepsy. It was also pointed out that there are currently too few confirmed studies to use BDNF as a marker for the development of epilepsy [41]. Aungst et al. showed that TrkB receptor activation and reactive axonal sprouting are critical factors in injury-induced hyperexcitability and may contribute to the neurological complications of traumatic brain injury [50].

On the other hand, Gu et al. have shed a different light on the development of posttraumatic epilepsy. Their results suggest that after damage to the cerebral cortex, the chronic activation of TrkB by a partial agonist of this receptor could enhance GABAergic inhibition and suppress posttraumatic epileptogenesis, which in turn could be used to prevent posttraumatic epilepsy and treat other diseases whose pathogenesis involves impaired interneuronal parvalbumin function [51]. The authors of animal model analyses tend to suggest that the models use phenomena that occur in different brain regions where BDNF/TrkB signaling may have the opposite effect, namely decreased excitability in the neocortex and increased excitability in the hippocampus [31]. Research on an animal model in which a normal brain was transformed into an epileptic brain showed that the transient inhibition of TrkB kinase initiated after status epilepticus (SE) was sufficient to prevent SE-induced TLE and associated anxiety-like behavior in this mouse model. Unfortunately, the global inhibition of TrkB signaling leads to the abrogation of the neuroprotective role of BDNF/TrkB signaling and increased degeneration of neurons, and more selective inhibitors are being sought. The phenomenon of the inhibition of phospholipase C-γ1-dependent signaling pathways, which mediates the epileptogenic consequences of TrkB activation, raises high hopes for the treatment of epilepsy [31]. The therapeutic role of BDNF in epilepsy was highlighted by Falcicchia et al., who used encapsulated cell biodelivery (ECB) with genetically modified human cells manipulated to release BDNF to treat rats with induced epilepsy. They demonstrated that the continuous release of BDNF in the epileptic hippocampus reduced the frequency of generalized seizures by more than 80%, improved cognitive performance, and reversed many histological changes associated with chronic epilepsy [52]. The conflicting research results regarding the effect of BDNF/TrkB signaling in epilepsy has led to further research in this area [53].

### 4.2. Depression

Depression is one of the most serious problems facing health care systems around the world. Approximately 4–16% of the world’s population suffers from major depressive disorder (MDD) and the lifetime risk of a depressive episode is 14–20% [54,55,56,57]. Despite advances in the treatment of this disorder, complete symptom relief is not achieved in approximately 20–40% of patients treated with commonly used antidepressants [7,54,58] Treatment-resistant depression (TRD) is diagnosed in patients in whom treatment with at least two different antidepressants taken at effective doses over an adequate period of time has failed to improve the patient’s clinical condition [54,59]. Numerous studies have shown that depression is closely related to the abnormal neuronal plasticity processes that occur in the prefrontal cortex and limbic systems such as the hippocampus and amygdala [7]. In addition, the volume of the hippocampus has been shown to decrease in patients with depression, but it is not clear whether this phenomenon is a consequence of depression or rather a predisposing factor for its development (i.e., a vulnerability marker or a consequence of this disorder) [60]. According to some authors, the reduction in hippocampal volume follows the neurotoxicity hypothesis, which states that long-term exposure to glucocorticoids increases the vulnerability of neurons to damage [61].

In recent years, as part of the search for better diagnostic and therapeutic methods for depression, numerous scientific papers have been produced linking mood disorders and antidepressant effects to BDNF and its receptor TrkB [20]. After several decades in which the development of depression was dominated by the “monoamine hypothesis,” according to which pathogenesis was based on decreased basal levels of serotonin, norepinephrine, and possibly dopamine, monoamine deficits might not reflect a core feature of the pathophysiology of depression and may instead represent the result of neural dysfunction [62,63].

The neurotrophin hypothesis was formulated in 1997 by Duman et al. and stipulates that MDD develops secondary to impaired neurogenesis in regions of the brain that regulate emotion and memory [64]. According to this hypothesis, impaired neurogenesis is a response to lower BDNF expression, and it predicts that antidepressants are effective because they increase BDNF expression, thereby solving the problem of impaired neuronal plasticity [20,62].

In postmortem studies of brain tissue, particularly from the hippocampus and amygdala, levels of BDNF mRNA and protein were reduced in people with depression [7,20]. In addition, decreased levels of TrkB and TrkB mRNA were found postmortem in these individuals. A reduction in BDNF levels was also observed in postmortem samples from patients who had committed suicide [7,65]. On the other hand, people who were treated with antidepressants in the period immediately before their death showed increased BDNF expression in the hippocampus [65,66]. Strikingly low levels of BDNF were found in the serum of patients suffering from MDD, and an increase in concentration was detected after they took “typical” antidepressants [67].

Despite enthusiastic past reports on the relationship between currently used antidepressants and BDNF, the initial hopes for BDNF-dependent signaling induced by antidepressants were questioned by later studies, and clinical practice still shows the need to search for new therapeutic directions [66,68]. Researchers presented both supportive and negative arguments side by side, and a comparison of these conflicting research results was made by Groves [65]. It was pointed out that decreased BDNF levels are not specific to mood disorders, and a similar decrease was also observed in schizophrenia or dementia [20].

In response to the neurotrophin hypothesis, meta-analyses were performed in the following years that confirmed the finding of low serum BDNF concentrations in untreated patients with depression and their normalization by antidepressant treatment, but the authors did not provide a definitive answer as to whether a low BDNF concentration should be treated as a trait or state marker [67,69,70]. The next meta-analysis, which studies a large group of subjects and controls (n = 9484), confirmed that serum BDNF concentrations are low in untreated depressed patients and normalized by treatment with antidepressants. It also noted a large amount of unexplained heterogeneity between studies in terms of outcomes, inadequately powered study designs, and publication bias that may have led the authors to make incorrect inquiries. In addition, no correlation was found between the BDNF concentration and depression symptom severity [62].

One study examined the association between the Val66Met polymorphism and MDD, with conflicting results. Verhagen et al. showed that in the total sample, BDNF-Val66Met polymorphism was not significantly associated with depression. However, sex-specific analyses revealed significant effects in both allelic and genotypic analyses in males [34]. In the meta-analysis by Ming Li et al., this SNP was not associated with the risk of MDD in both European and Asian populations, which was found in several previous studies [38]. Thus, the Met allele of rs6265 is associated with poorer episodic memory performance, which is often observed in MDD, while the Vall allele is associated with a higher trait anxiety and higher mean neuroticism score, which is also considered a potential factor in the development of depression [34]. In turn, Zhao et al. showed that carriers of the Met allele were significantly more likely to develop depression when exposed to stress in both childhood and adulthood [35].

Based on a small review, Kishi et al. concluded that BDNF should be considered as a biomarker of MDD state and treatment response rather than a risk factor for MDD. They point out that several genome-wide association studies (GWAS) of MDD have not found markers associated with a clinical risk of MDD, which may be related to the substantial heterogeneity of the pathophysiology of depression [40]. Moreover, both serum and plasma levels of BDNF were lower in patients with MDD than in the control group and increased after drug treatment and electroconvulsive therapy. A positive effect of Val66Met polymorphism on treatment response was demonstrated, but no difference in the frequency of this SNP was found between patients with MDD and the control group [40]. Patients who are heterozygous for Val66Met respond better to antidepressant treatment than homozygous patients who have either the Val or Met allele [71].

Conventional antidepressants based on monoaminergic neurotransmission have two fundamental limitations: First, their onset of action is delayed by up to months; and second, up to one-third of patients, some of whom have TRD, do not respond adequately to treatment [72]. Although BDNF and TrkB ligands cross the blood–brain barrier (BBB), the problem arises that they exert opposite effects in different cell types and brain regions [73,74]. For example, BDNF and TrkB agonists have antidepressant effects in the hippocampus but prodepressant effects in the nucleus accumbens. Furthermore, the peripheral infusion of BDNF causes hyperalgesia, which is an additional limitation to the use of BDNF in the form of infusions [74,75].

In recent years, there has been hope regarding treating TRD with ketamine, a noncompetitive N-methyl-D-aspartate (NMDA) glutamate receptor antagonist that intranasally activates BDNF-TRkB signaling directly in the brain [57,72]. In research, its racemic form, consisting of (S)-ketamine and (R)-ketamine, is also used as an intravenous infusion. Daly et al. showed in their randomized clinical trial that, when compared with treatment with antidepressants and placebo, (S)-ketamine nasal spray and antidepressant treatment reduced the risk of relapse by 51% in patients who achieved stable remission and by 70% in those who achieved stable response [59].

The greatest advantage of ketamine is undoubtedly its rapid action, which is difficult to achieve with the use of conventional antidepressants. It has been shown that after the intravenous administration of ketamine, the risk of suicidal ideation decreases within one day of ingestion, and the effect lasts up to one week [57]. Ketamine is thought to enhance brain plasticity by stimulating BDNF production and activating the mammalian target of rapamycin (mTOR). Ketamine activates amino-hydroxy-methyl-isoxazolepropionic acid receptors (AMPA receptors) via NMDA receptors, which may modulate downstream signaling pathways that appear to decrease the phosphorylation of eukaryotic elongation factor 2-kinase and lead to increased BDNF production [22,23]. Furthermore, it is assumed that BDNF production is additionally enhanced by the downstream modulation of mTOR, leading to increased brain plasticity through dendritic growth and enhanced synaptic transmission [22,24].

There are numerous reports linking BDNF concentrations in serum and plasma to electroconvulsive therapy (ECT); however, they are not conclusive [76]. Probably the largest meta-analysis on this topic to date was performed by Pelosof, Brunoni et al., which included 28 studies (n = 778) and showed that BDNF levels increase significantly after ECT treatment but with a small effect size [77]. With such a large number of studies, the meta-analysis was characterized by a large heterogeneity that could bias the conclusions. Nevertheless, subgroup analyses revealed a significant increase in BDNF levels when measured in plasma but not in serum. On the one hand, moderator analyses showed that a greater increase in BDNF concentration in blood was detected in the group of younger patients; on the other hand, plasma determinations were more frequent than serum determinations in younger patients (43 years old vs. 52 years old). Moreover, the authors pointed out that ECT has a more significant effect on the resolution of depression symptoms than on changes in BDNF concentration. A previous meta-analysis by Brunoni et al. involving 11 studies (n = 221) showed that BDNF blood levels increased after treatment, with a net effect similar to that observed for antidepressants [78]. Sorri et al. conducted studies in a small group (36 patients), demonstrating that ECT did not cause significant changes in serum BDNF levels and that plasma BDNF levels decreased between baseline and 2-h samples during the fifth ECT session. The authors noted the limitations of their small patient population, but they emphasized that although many studies confirm the increase in serum BDNF levels after ECT, there is also a large group of studies confirming their observations that reported no effect of ECT on serum or plasma BDNF levels during or after the ECT series. According to Sorri et al., one of the reasons for this could be the different mechanisms of therapeutic action of ECT and antidepressant drugs, since the effect of ECT mainly involves signaling systems other than serotonin and thus exerts less influence on BDNF levels [76].

There is another method that appears to be effective in treating refractory depression, bipolar affective disorder, and schizophrenia, but its mechanism of action is unclear, namely repetitive transcranial magnetic stimulation (rTMS). A number of researchers have reported an increase in blood BDNF levels in response to repeated TMS sessions, but there have been other studies suggesting that TMS has no effect on BDNF, so further research on this matter is needed [55].

### 4.3. Hunger Control Disorders

Anorexia nervosa (AN) is a severe mental disorder with a lifetime prevalence of up to 4% in women and 0.3% in men, with the risk of death being increased five times or more [79,80]. Only one-third to one-half of female patients remain weight stable [81]. Even in patients who initially had complete remission, relapse is common, and of all mental illnesses, anorexia has the highest mortality rate [21,81]. On the other hand, obesity, with its consequences, such as type 2 diabetes or cardiovascular diabetes, is a growing problem worldwide. The worldwide prevalence of overweight and obesity has doubled since 1980, so that today almost one third of the world population is classified as overweight or obese. The problem of increasing obesity affects all populations, but especially the elderly and women [82]. In the pediatric population, obesity is a complex problem that affects children of all ages. As in adults, one-third of children and adolescents in the United States are classified as overweight or obese [83].

Appetite is regulated by a number of hormones, the most prominent of which are growth hormone secretagogues such as ghrelin, which exhibit orexigenic activity, and anorexigenic hormones such as peptide YY (PYY) or BDNF [84].

The hypothalmus and especially the dorsal vagal complex (DVC) are involved in the regulation of the feeding process. In addition, the mesolimbic dopamine reward pathway is involved in hedonic food intake [85,86]. The main appetite-regulating organ is the nucleus arcuatus of the hypothalamus (ARC), which according to some reports is partially devoid off the blood–brain barrier and therefore receives signals involving substances circulating in the blood [87]. The other hypothalamic structures involved in the regulation of food intake are the paraventricular nucleus (PVN), ventromedial hypothalamus (VMH), dorsomedial hypothalamus (DMH), and lateral hypothalamus (LH). BDNF is synthesized in the VMH, DMH, LH, and PVN, but not in the ARC. However, this does not mean that the ACR has no BDNF effects; TrkB has been detected in ACR receptors and BDNF-containing nerve fibers. The BDNF protein concentration in the DVC has been shown to be decreased by fasting and increased by refeeding [86].

The results of preclinical studies in rodents and mice suggest that exogenous BDNF intake is associated with weight loss, whereas animals that have reduced BDNF expression tend to be obese and insulin resistant [88,89,90]. In turn, Liao et al. postulate that leptin and insulin induce the local translation of BDNF mRNA in dendrites of hypothalmic neurons. Their results show that mutations in the BDNF gene lead to leptin resistance, which is a major risk factor for obesity, despite the normal activation of leptin receptors. Liao et al. argue that the efficient action of leptin on a specific area of the CNS requires undisturbed transmission with the involvement of BDNF in this area [91].

Studies in humans also confirm the effects of the BDNF/TrkB pathway on the regulation of the feeding process. Cases have been described of patients with de novo missense mutation in the TrkB gene who exhibit severe obesity in addition to complex developmental syndrome, impaired learning and memory, and impaired nociception. The mutation of NTRK2, which encodes TrkB, appears to lead to a unique human syndrome of hyperphagic obesity, and researchers have emphasized that haploinsufficiency in the above gene is sufficient to develop the obesity phenotype in both humans and mice [92,93]. Such observations also concern patients with WAGR syndrome, which consists of Wilm’s tumor, aniridia, genitourinary abnormalities, and mental retardation and is a consequence of large truncations within chromosome 11 containing the human BDNF gene. In patients with WAGR syndrome in which the mutation had affected the BDNF gene, 100% of patients were obese, whereas only 20% of patients with WAGR syndrome with intact BDNF alleles developed obesity [86]. There are studies linking the Val66Met polymorphism, which affects BDNF-mediated signaling, to an increased body mass index (BMI) [36,94]. On the other hand, certain researchers have demonstrated that this polymorphism is related to anorexia nervosa; however, this has not been confirmed and is not justified based on the meta-analysis performed by Brandys et al., among others [36].

Despite numerous reports linking obesity to mutations in the BDNF gene or its receptor, and despite the demonstrated association between obesity and a reduction in BDNF protein in the hypothalamus, a meta-analysis (307 obese patients and 236 controls, 517 subjects) showed that circulating BDNF levels in both plasma and serum did not differ between obese patients and the control group, implying that the previously hypothesized use of exogenous BDNF to prevent the deleterious effects of metabolic syndrome needs to be revisited [95].

Another study that examined the relationship between overweight/obesity and executive control (EC) in young adults and further analyzed the mediating effect of neurotrophic factor (BDNF) showed that overweight or obesity is related to lower BDNF levels, which in turn is related to lower EC [96].

A significant amount of research links BDNF to anorexia nervosa. BDNF is thought to exert an anorexigenic effect, but it remains unclear how serum BDNF levels correlate with the onset and course of anorexia nervosa. A meta-analysis (AN n = 155, healthy controls n = 174) showed that AN is associated with decreased serum BDNF concentrations compared with healthy controls and that the increase in BMI during the treatment of AN is related to the growth of BDNF concentrations. In addition, the study pointed out the comorbidity between AN and depression and the fact that antidepressant treatment normalizes BDNF levels in patients with isolated depression, whereas this treatment has little or no efficacy in AN. This may be due to the fact that stress is the fundamental element in the development of depression (the core component), while the inability to cope with stress is only one component of AN, which is associated with an altered mechanism of reward processing and an increased vulnerability to anxiety and anhedonia, which in turn leads to the development of the phenomenon of “hunger addiction” [21].

Another meta-analysis also confirmed decreased BDNF levels in patients with AN compared with the control group, whereas BDNF levels were slightly higher in recovered AN patients. However, after subdivision of patients into “binge-purge” AN (AN-BP) and “restricting” AN (AN-R), only the latter group showed a confirmed, statistically significant association with decreased BDNF concentration and could be considered a state marker for AN-R, with the authors leaning toward the thesis that a decreased BDNF concentration is a response to starvation. In addition, the authors emphasized that there was still no clear evidence as to whether changes in BDNF levels exacerbate AN or whether the expressions of adaptive mechanisms are aimed, for example, at reducing resting brain energy expenditure. The low BDNF concentration could have been the result of proinflammatory cytokines such as TNF-alpha inhibiting the plasminogen activator-dependent cleavage of pro-BDNF into BDNF, as when body weight is restored, the inflammatory process calms down and BDNF concentration increases again. In addition, an increase in the BDNF concentration improves neuroplasticity and cognitive functions, which improves the treatment process, especially in the field of psychotherapy. Unfortunately, there is a concern regarding a possible anorexigenic effect of BDNF, which could perpetuate poor appetite in patients with AN if BDNF levels increase; however, according to the authors of the above study, these mechanisms still need to be better understood and researched [97]. In an attempt to explain low BDNF concentrations as well as high ghrelin concentrations in AN patients, it was hypothesized that these may be adaptive mechanisms aimed at promoting food intake during chronic hunger [84,85].

An important role in the development of AN may be that hormonal signaling is disrupted. A study by Mancuso et al. showed that patients with AN have lower BDNF concentrations before a meal but have a greater increase 120 min after food consumption than the control group. This could be responsible for the greater suppression of homeostatic and hedonistic appetite in AN, but further studies are needed to confirm this relationship [84].

One study conducted on overweight or obese patients and a healthy control group and found that dieting combined with exercise significantly reduced circulating BDNF levels in women, while exercise alone reduced circulating BDNF levels in men (with only a trend shown in women), with these findings being essential for interpreting the results of AN patients in the context of excessive exercise [26].

### 4.4. Schizophrenia

Schizophrenia (SCZ), despite its relatively low prevalence, which is 0.32% of the world population (0.45% in adults) according to the most recent data from WHO, is a disease with a significant burden of disease, expressed in terms of the DALY index (disability-adjusted life-years), among others [98]. Despite advances in treatment that followed the widespread use of second-generation atypical antipsychotics, outcomes remain suboptimal. A systematic review reported that the median proportion of schizophrenia patients who met criteria for clinical and social recovery was only 13.5%, while partial remission was achieved in less than 40% of patients [9,98,99].

BDNF plays a role in many processes confirmed in the pathophysiology of schizophrenia, including the survival and plasticity of dopaminergic, cholinergic, and serotonergic neurons. It is postulated that the structural disorganization of networks occurs in patients with schizophrenia during brain development and that this may later contribute to the disease and explain some of the morphological, neurochemical, and cytoarchitectural abnormalities found in the brains of patients with schizophrenia [100].

There are conflicting reports on the relationship between BDNF serum levels in patients with schizophrenia. According to most researchers, BDNF levels are lower in chronically treated SCZ patients than in control groups, but the type of treatment is a factor that may affect the correct interpretation of the results [9]. The first systematic review with a meta-analysis that investigated the dependence of BDNF blood levels in SCZ compared to healthy controls found that the BDNF blood level is decreased in drug-treated and non-drug-treated patients with schizophrenia, regardless of sex, and that this effect increases with age and is independent of drug dose. It should be noted, however, that the study featured an unexplained heterogeneity in terms of the results. In the analysis performed, the majority of studies confirmed reduced peripheral BDNF levels in schizophrenia, whereas two studies reported higher serum BDNF concentrations or serum protein in schizophrenia compared with healthy controls. Only two assays showed no significant differences in plasma BDNF levels in non-medicated SCZ patients compared to controls, with increased BDNF levels observed in the same subjects after antipsychotic treatment [101].

Most research has focused on the impairment of cognitive dependence in patients with SCZ and the effect this has on BDNF levels. Cognitive deficits, which include decreased processing speeds, verbal memory, and executive functions, as well as decreased learning and memory, are common in patients with SCZ from the first episode of the disease [102].

Neurocognitive deficits have been described in relation to the Val66Met polymorphism, and attempts have been made to associate its occurrence with a tendency to develop SCZ [102,103]. One of a number of studies using magnetic resonance imaging investigated the association between BDNF, the Val66Met SNP, and progressive brain volume changes in patients with recently diagnosed schizophrenia, and Met allele carriers were found to have a significantly greater reduction in frontal gray matter (GM) than Val homozygous patients [42]. In addition, regardless of genotype, it was shown that patients exposed to higher doses of antipsychotics during the 3-year follow-up period also had a greater reduction in GM, and although this could be explained by the fact that in patients with more severe disease symptoms, expecting greater reductions in brain volume required higher doses of medication, studies in monkeys also show that exposure to antipsychotics reduces fresh brain weight and volume by ~10%, independent of treatment with haloperidol or olanzapine [42].

Ahmed et al. performed a meta-analysis, and the results indicate significantly lower right and left hippocampal volumes in patients with schizophrenia carrying the Met allele as well as a reduction in the frontal volume of these patients that did not reach statistical significance in the meta-analysis. The second part of the study also showed that lower BDNF levels correspond to lower volumes in terms of the right and left hippocampus [103].

In patients with schizophrenia who received cognitive remediation therapy (CRT), the volume of the hippocampus increased compared with patients who received treatment as usual, which the authors explained by the possible influence of BDNF, whose increase in concentration during the course of CRT was demonstrated in another study [104,105].

ECT is an alternative treatment for schizophrenia and is reserved for patients who do not respond to standard pharmacological treatment. In Western countries, ECT is used in 2.9–36% of patients with schizophrenia, while in China it is a common treatment method used in up to 55.2% of patients [106]. Studies in rats mostly show a significant increase in BDNF transcripts in response to electroconvulsive seizures (ECS), but this effect varies with the brain region and number of repeated sessions [9]. A recent cohort study including 60 patients with resistant schizophrenia showed that patients receiving ECT (n = 45) had increased BDNF levels after treatment compared with patients receiving conventional pharmacological treatment (usual treatment with antipsychotics) (n = 15), with a statistically significant difference compared with their baseline BDNF level. Pharmacologically treated patients, on the other hand, did not show a significant increase in BDNF concentration. Moreover, patients treated with ECT showed a better response [107]. Li et al. redefined BDNF as a good marker of response to antipsychotic treatment (AP) and AP in conjunction with ECT [108]. In a meta-analysis of 11 RCTs (n = 1018), ECT was shown to impair memory, and therefore the early detection of these disorders is necessary to prevent long-term cognitive impairment [106].

### 4.5. Alzheimer’s Disease

With an aging population, Alzheimer’s disease (AD) poses a major challenge to the functioning of health care systems. It is estimated that there are more than 50 million dementia patients worldwide, with 22% of those over 50 years of age suffering from AD dementia, AD prodromal dementia, and AD preclinical dementia [109].

The pathogenesis of AD is multifactorial, but the accumulation of β-amyloid (Aβ) deposits in the brain in the form of senile plaques and over-phosphorylated tau protein as neurofibrillary tangles plays a key role [110,111]. For many years, Aβ accumulation was believed to trigger and accelerate disease progression but to act independently of the tau protein; however, recent observations suggest that the two processes are interdependent and act synergistically [110].

Busche et al., in their thesis on the synergistic effect of Aβ and tau protein (Aβ-tau synergy) in the pathogenesis of AD, state that at present it is not clear in which mechanism these interactions take place. In their considerations, they do not mention the possible role of BDNF, which we believe should be considered [110].

The role of BDNF in the development of AD appears to be multidirectional. It is known that BDNF depletion is associated with increased tau phosphorylation, Aβ accumulation, neuroinflammation, and neuronal apoptosis [111]. The above changes are the result of the involvement of BDNF in downstream signaling pathways such as mitogen-activated protein kinase/extracellular signal-regulated protein kinase (MAPK/ERK), PI3K, and phospholipase Cγ/protein kinase C (PLCγ/PKC), which are associated with the activation of the transcription factor cAMP-response element binding protein (CREB), which is critical for synaptic plasticity [111].

The involvement of BDNF in the development of AD was confirmed by a small number of meta-analyses. Two independent meta-analyses by Qin et al. and Ng et al. showed that AD patients had lower BDNF blood levels compared to the control group [112,113]. In turn, Due et al. demonstrated the downregulation of BDNF in blood, cerebrospinal fluid, hippocampus, and cortex in AD patients [114].

From a clinical point of view, there is a need to find an AD biomarker that is sensitive, for the purpose of diagnosis in the early stages of the disease, before clinical symptoms appear, but which is also rapid and non-invasive, which is why BDNF has attracted the attention of researchers [3]. These hopes were disappointed by Ng et al. because, although they demonstrated that the decrease in peripheral BDNF concentration is associated with AD, this phenomenon occurs only in the advanced stage of the disease, which is most likely related to the earlier increase in BDNF secretion as part of the compensatory and neuroprotective mechanism, and only the exhaustion of these possibilities leads to a decrease in BDNF levels in the blood [113].

The use of BDNF as a marker in the detection of AD or mild cognitive impairment (MCI), which is a transitional state between normal aging and dementia, was also disputed by Xie et al. in their meta-analysis. The authors also showed that AD is associated with decreased levels of peripheral BDNF; however, ROC curve analysis showed that peripheral BDNF levels may not be an optimal biomarker for AD or MCI diagnosis due to the lower AUC, lower sensitivity, and poor specificity [115].

Given that BDNF has such a significant association with the development of AD, it became clear that therapies targeting BDNF-related mechanisms needed to be sought [116]. Most symptomatic drugs approved for the treatment of AD affect blood BDNF levels. This is true for both acetylcholinesterase inhibitors (AChEI) and a select N-methyl-d-aspartate receptor (NMDAR) antagonist [111]. To date, despite controversy, the FDA has approved the first and only disease-modifying drug in AD therapy, the IgG1 anti-Aβ monoclonal antibody (aducanumab), which reduces accumulated Aβ, but its use does not result in clinical benefits, including an improvement in cognitive function [111,117].

Given the theory of Busche et al., which points to the synergy of Aβ-tau, it may turn out that therapies combining anti-Aβ monoclonal antibodies with anti-tau therapies will be used in the future [110]. In our opinion, the effect of such treatment on BDNF should then also be investigated, because to date there is only isolated evidence in the literature of a link between BDNF and anti-Aβ monoclonal antibodies.

Such multi-drug therapy was also postulated by Numakawa et al., who investigated the effect of flavonoids on neuroprotection and also noted that a promising direction in AD therapy is the inclusion of BDNF as a promising target for AD in combination with Aβ/tau-targeted therapy [118].

Among the many directions in AD therapy targeting BDNF action, attention is focused on drugs acting on microRNA (miRNA), i.e., endogenous non-coding RNA that regulates gene expression [111,119,120].

The disruption of miRNA is closely related to the pathophysiology of neurodegenerative diseases, but depending on the type of miRNA we are dealing with, it may either inhibit the development of the disease or be a factor that exacerbates it [120]. Therefore, one part of the strategy could focus on the development of a therapy based on the stimulation of neuronal miRNA expression, while another part could aim at its selective inhibition. Currently, most research in this area is still at an early stage and needs further validation [120,121].

### 4.6. Therapeutic Properties of BDNF

Due to the numerous effects of BDNF on nervous system disorders, attention is being drawn to the potential of its therapeutic use. Unfortunately, there remain certain significant limitations to such therapy, namely the short half-life and side effects of BDNF as well as, according to some authors, the need to cross the blood–brain barrier (BBB) [111]. Among the agents tested, there are those that increase the transcription, translation, and secretion of BDNF, with these including gene therapy using viruses as vectors, the administration of plasmid DNA encoding BDNF using nanoparticles, or the use of ECB employed in the treatment of epilepsy with genetically modified human cells designed to release BDNF [5,10,52,122,123,124]. Another strategy could be to administer BDNF from an exogenous source via the intranasal route, thereby bypassing the BBB, but this approach also has its limitations, as nasal mucus hinders the absorption efficiency [5,10,41]. In addition, many sources indicate that BDNF has the ability to overcome the BBB, so the extent to which such delivery enhances the penetration of exogenous BDNF into the CNS should be determined [73,103]. Efforts are also being made to increase TrkB receptor activity. For this purpose, TrkB-FL (full-length tropomyosin related kinase B) agonists, BDNF mimetics, TrkB-Fl transactivators, or mediators of TrkB-FL-mediated effects such as adenosine A2A receptor agonists have been used. Unfortunately, there are also limits to the efficacy of TrkB receptor-mediated effects, and the phenomenon of endocytosis plays a key role [5].

Moreover, the use of drugs that affect BDNF/TrkB signaling is controversial because there is still a lack of clear, repeatable test results in most diseases, and even large meta-analyses often fail to provide satisfactory answers to important questions that would allow for the design of reliable randomized trials.

## 5. Conclusions

The fact that BDNF, despite its suggestive name, is secreted in large amounts from tissues and peripheral organs outside the CNS and is stored in the blood in a relatively large amount in platelets means that disturbances in its concentration can occur in various diseases, including those not directly related to neuropsychiatric disorders, making the interpretation of results much more difficult, especially in patients with multiple pathologies. In the near future, it may turn out that the most significant benefit that comes from evaluating its concentration is related to the prediction of the course of the disease and the effectiveness of its treatment.

A better understanding of the disease processes that occur through different pathways involving BDNF could be useful for the development of drugs with new mechanisms of action which use BDNF analogues, agonists, or antagonists for its receptors. However, given the divergence of the current state of research, this process could take many years.

## 6. Methodology

To describe the topic addressed in the title of the article, the authors searched PubMed and the Cochrane Library using terms related to the topic, such as “anorexia nervosa BDNF,” “schizophrenia BDNF,” “depression BDNF,” “Alzheimer disease BDNF,” while also trying to mix these keywords with other keywords such as “treatment,” “novelty.” The search was conducted throughout the month of August 2022. Based on the authors’ clinical and research experience, 124 articles were selected that were directly related to the topic of this review. Of these, 115 citations were published after 2003. With one exception, only English-language articles published in peer-reviewed journals were selected. The authors wish to point out that this is a narrative review. Therefore, guidelines suggesting exceptional measures for article searches (e.g., Preferred Reporting Items for Systematic Reviews and Meta-Analyses) were not used.

## Figures and Tables

**Figure 1 brainsci-13-00163-f001:**
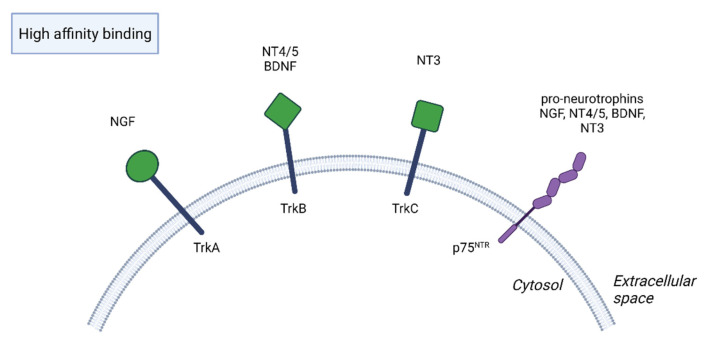
NGF, nerve growth factor. NT4/5, neurotrophin 4/5. BDNF, brain-derived neurotrophic factor. NT3, neurotrophin 3. Immature forms of neurotrophins bind with high affinity to p75 neurotrophin receptor (p75NTR) and with low affinity to tropomyosin-related kinase Trk receptors, while mature forms have a reverse binding behavior.

**Figure 2 brainsci-13-00163-f002:**
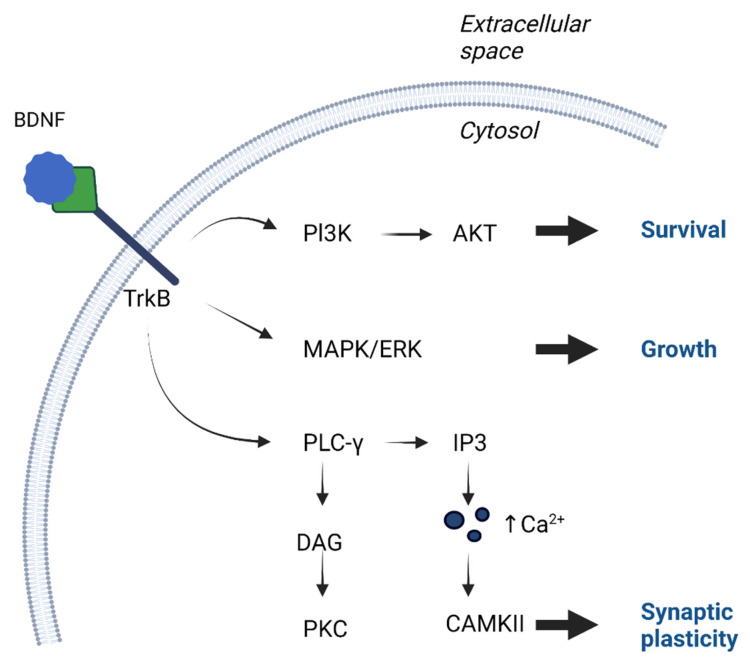
Activation of TrkB receptors occurs through phosphorylation, which leads to downstream pathways activation. Cell growth occurs mostly through activation of mitogen-activated protein kinase (MAPK) or extracellular signal related kinase (ERK). Cell survival is possible through phosphatidylinositol 3-kinase (PI3-K), which activates the serine/threonine kinase AKT. Greatest influence on synaptic plasticity is phospholipase C gamma (PLC-γ) pathway, which activates inositol trisphosphate (IP3) receptor, which activates calcium/calmodulin-dependent kinase. Moreover PLC-γ, with the participation of diacylglycerol (DAG), activates protein kinase C (PKC).

## Data Availability

Data available in a publicly accessible repository. The data presented in this study is available by accessing citations below.

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
