# Peer review of "The Role of Brain-Derived Neurotrophic Factor (BDNF) in Diagnosis and Treatment of Epilepsy, Depression, Schizophrenia, Anorexia Nervosa and Alzheimer’s Disease as Highly Drug-Resistant Diseases: A Narrative Review"

_brainsci, 2023, doi:10.3390/brainsci13020163_

Round 1

Reviewer 1 Report

The manuscript deals with “The role of Brain-Derived Neurotrophic Factor (BDNF) in diagnosis and treatment of epilepsy, depression, schizophrenia and anorexia nervosa as highly drug-resistant diseases: a narrative review”. I believe that this study can be published but only after considering some comments: 

1. The introduction section is very long and the purpose of writing this manuscript is unclear.

2. What is the relationship between the neurotransmitter GABA and BDNF between neurons?

3. According to the many mentioned reports about BDNF changes in epilepsy, what can we conclude in general?

4. It is better to also investigate Alzheimer's and Parkinson's disease.

5. Considering the changes in neurotransmitters such as dopamine, serotonin, acetylcholine, etc. in the studied diseases, can we find a relationship between the alterations in these neurotransmitters and BDNF levels?

6. Has the methodology been done for this study? Did the authors perform a meta-analysis?

7. Many of the mentioned sentences and contents are without references.

Reviewer 2 Report

I'd like to thank the authors for the opportunity to review the paper. They seem to have done a mammoth job in combing through the BDNF literature! In general, I think the effort has merit, but it is in need of pruning and some thinking about what they hope their contribution will be. I am, for example, not sure what the review adds in terms of depression or schizophrenia when read besides the excellent work by Arosio et al. and Nieto et al., both from last year or that by Xu et al., from 2020, or for epilepsy when compared to Wang & Zhong 2021 and the very comprehensive systematic review by Iughetti et al. (2018). What does this very long but pretty unstructured narrative review add that recent, more specialised systematic reviews hasn't already pointed out?

Some assorted remarks:

I did find it a bit surprising that there was no mentioning of Alzheimers apart from the gene findings, not even in the treatment section considering that there's a clinical trial now recruiting (https://www.clinicaltrials.gov/ct2/show/NCT05040217) that uses an adenovirus vector. Perhaps this should be added.

I would advise against discussing pre-clinical data expect perhaps in introductory headings ('several studies in rodents indicate a possible role for BDNF in this-or-that [XX-YY]', not mammoth paragraphs such as 482-497. The scope is disorders and their treatment, and there is a jungle of often conflicting preclinical findings out there.

Regarding anorexia nervosa - the Keeler meta-analysis interprets their findings as lowered BDNF being a state marker related to starvation in itself, not necessarily to the pathogenesis. Should probably be mentioned.

As the relevance of BDNF for the pathogenesis of any of the disorders discussed it'd be better if that is reflected in a more careful language throughout the review, e.g. at line 99 it'd be nice with a 'possible' inserted before 'role'.

Line 199: "During the course of research, it was discovered that BDNF may have neuroprotective effects as well as its overexpression may contribute to the development of certain diseases, including epilepsy" Rather redundant sentence, but citation for the neuroprotection? 

Regarding depression: standard antidepressants display a significant effect on core symptoms already after a week (e.g. Hieronymus et al., 2016) (albeit a weak one), so the wording should probably be different here.

References are often missing, but sometimes given in abundance where it's unnecessary. E.g. "Epilepsy is a chronic disorder of the central nervous system that affects an estimated 1% of the world's population. Except for a few established causes, the origin of epilepsy in the majority of patients cannot be determined by currently available diagnostic methods." at lines 227-229 but "Depression is one of the most serious problems facing health care systems around 314 the world. Approximately 4-16% of the world's population suffers from major depressive 315 disorder (MDD) and the lifetime risk of a depressive episode is 14-20% [46–49] ." at lines 314-138; I don't see why the last paragraph warrants four references but the first none. Indeed, I don't see why the paragraphs needs to be there in the first place; statements about the prevalence of this-and-that is, to my mind, redundant; the review is very long and would benefit from pruning - lines 455 to 466 are for example also largely redundant, in my mind.

"This section may be divided by subheadings. It should provide a concise and precise description of the experimental results, their interpretation, as well as the experimental conclusions that can be drawn" is left at lines 452-454 and 574-576.

Reviewer 3 Report

The review “The role of Brain Derived Neurotrophic Factor (BDNF) in diagnosis and treatment of epilepsy, depression, schizophrenia and anorexia nervosa as highly drug-resistant diseases: a narrative review” by Gliwińska and co-authors is focused on the role of BDNF in certain types of neurological and psychiatric diseases.

The review encompasses the main trends in the field and presents a comprehensive view on the topic.

The manuscript has some minor and major problems.

1.     “Neurotrophins are synthesized as immature forms that have high affinity for the p75 neurotrophin receptor (p75NTR) and low affinity for tropomyosin-related kinase (Trk) receptors. They are then converted by proteolysis into mature forms of neurotrophins that act through Trk receptors, although they also have low affinity for p75NTR.” – the statement needs to be supported by citation.

2.     TrkB or trkB? Authors should use a uniform spelling form.

3.     The authors should explain at the beginning of the review what is meant by synaptic plasticity.

4.     The authors should include a vast majority of data on the regulatory small RNAs and microRNAs, which is highly important.

5.     Other studies in people with schizophrenia found that carriers of the Met allele had lower peripheral blood BDNF concentrations [2,38] , which contradicts other studies that found no differences between Met carriers and Val/Val homozygotes in the measurement  of peripheral BDNF in plasma and serum [28] . - What is meant by peripheral blood BDNF concentrations?

6.     One of the major problems of the paper is English academic writing, the authors need to perform an accurate proof reading by a native speaker or use a specialized service.

Please, find below the sentences which require revision.

7.     “Among the aforementioned neurotrophins, BDNF has received the most attention 98 because of its role in the pathogenesis of numerous neurological and psychiatric disorders 99 and because of its potential therapeutic value, but because of its lack of specificity, its util-100 ity may be limited to monitoring response to treatment without the possibility of using it 101 as a predictive marker of disease occurrence.” - It’s bad English.

8.     “Their results suggest that after damage to the cerebral cortex, chronic activation of TrkB by a partial agonist of this receptor may enhance GABAergic inhibition and suppress posttraumatic epileptogenesis, which in turn could be used to prevent posttraumatic epilepsy. and in the treatment of other diseases whose pathogenesis involves impaired parvalbumin interneuronal function [45] . The authors of the analysis on animal models tend to suggest that the models used phenomena occurring in different regions of the brain where BDNF/TrkB signaling may have the opposite effect, namely reduced excitability in the neocortex and increased excitability in the hippocampus [31] .” -  Something is wrong here.

9.     “The initial hopes for BDNF and its effect by common antidepressants were chal-352 lenged by subsequent studies [56] .” -  Something is wrong here.

10.  “Despite numerous reports linking obesity to mutations in the BDNF gene or its receptor, and despite the demonstrated association between obesity and a reduction in BDNF protein? in the hypothalamus, a meta-analysis (307 obese patients and 236 control 517 subjects) has shown,…..” - .” -  Something is wrong here.

11.  The authors should provide a more recent citation on anorexia statistics, reference 68 is too old.

12.  “Inhibition of the phospholipase-dependent C-γ 1 (PLC γ 1) pathway, which mediates the epileptogenic consequences of TrkB activation [31] .” – It is not clear what is meant here.

13.  “Treatment-resistant depression (TRD) is diagnosed in patients in whom treatment with 319 two different antidepressants at appropriate doses and for an appropriate period of time 320 has failed to improve the patient's clinical condition [46,51] .” – bad English.

14.  “This section may be divided by subheadings. It should provide a concise and precise 452-454. description of the experimental results, their interpretation, as well as the experimental 453 conclusions that can be drawn.” – these sentences are repeated twice in the review – lines 452-454 and 574-576.

15.  “……..dose, but it should be noted that the study  was burdened with unexplained heterogeneity in study results.” – bad English

16.  “The first systematic review with meta-analysis that investigated the dependence of BDNF blood level in SCC compared to healthy controls found that BDNF blood level is decreased in drug-treated and…….” – what is SCC?

17.  “There are also attempts to increase TrkB receptor activity, for this purpose TrkB- FL (full length tropomyosin related kinase  B) agonists, BDNF mimetics, TrkB -Fl transactivators or mediators of TrkB - FL -mediated effects such as adenosine A2A receptor agonists .” -  Something is wrong here.

In addition, lines need revision: 66, 73, 94, 108, 119, 126, 136, 191, 268-269, 446, 543-546.

Major revision is needed

Round 2

Reviewer 3 Report

Accept in the present form. Corrections were made.